# StarDist Image Segmentation Improves Circulating Tumor Cell Detection

**DOI:** 10.3390/cancers14122916

**Published:** 2022-06-13

**Authors:** Michiel Stevens, Afroditi Nanou, Leon W. M. M. Terstappen, Christiane Driemel, Nikolas H. Stoecklein, Frank A. W. Coumans

**Affiliations:** 1Medical Cell Biophysics Group, Techmed Center, Faculty of Science and Technology, University of Twente, 7500 AE Enschede, The Netherlands; m.stevens@utwente.nl (M.S.); a.nanou@utwente.nl (A.N.); l.w.m.m.terstappen@utwente.nl (L.W.M.M.T.); 2General, Visceral and Pediatric Surgery, University Hospital and Medical Faculty, Heinrich-Heine University Düsseldorf, 40225 Düsseldorf, Germany; christiane.driemel@med.uni-duesseldorf.de (C.D.); nikolas.stoecklein@med.uni-duesseldorf.de (N.H.S.)

**Keywords:** circulating tumor cell (CTC), diagnostic leukapheresis (DLA), image segmentation, StarDist, CellSearch, ACCEPT

## Abstract

**Simple Summary:**

Automated enumeration of circulating tumor cells (CTC) from immunofluorescence images starts with a selection of areas containing potential CTC. The CellSearch system has a built-in selection algorithm that has been observed to fail in samples with high cell density, thereby underestimating the true CTC load. We evaluated the deep learning method StarDist for the selection of possible CTC. In whole blood sample images, StarDist recovered 99.95% of CTC detected by CellSearch and segmented 10% additional CTC. In diagnostic leukapheresis (DLA) samples, StarDist segmented 20% additional CTC and performed well, whereas CellSearch had serious failures in 9% of samples.

**Abstract:**

After a CellSearch-processed circulating tumor cell (CTC) sample is imaged, a segmentation algorithm selects nucleic acid positive (DAPI+), cytokeratin-phycoerythrin expressing (CK-PE+) events for further review by an operator. Failures in this segmentation can result in missed CTCs. The CellSearch segmentation algorithm was not designed to handle samples with high cell density, such as diagnostic leukapheresis (DLA) samples. Here, we evaluate deep-learning-based segmentation method StarDist as an alternative to the CellSearch segmentation. CellSearch image archives from 533 whole blood samples and 601 DLA samples were segmented using CellSearch and StarDist and inspected visually. In 442 blood samples from cancer patients, StarDist segmented 99.95% of CTC segmented by CellSearch, produced good outlines for 98.3% of these CTC, and segmented 10% more CTC than CellSearch. Visual inspection of the segmentations of DLA images showed that StarDist continues to perform well when the cell density is very high, whereas CellSearch failed and generated extremely large segmentations (up to 52% of the sample surface). Moreover, in a detailed examination of seven DLA samples, StarDist segmented 20% more CTC than CellSearch. Segmentation is a critical first step for CTC enumeration in dense samples and StarDist segmentation convincingly outperformed CellSearch segmentation.

## 1. Introduction

The circulating tumor cell (CTC) load detected by the CellSearch system reflects the state of disease [1,2]. Accurate enumeration of CTC is important for its use as a biomarker in patient risk assessment and to evaluate treatment response [3,4] or disease progression [5,6] during the course of the disease by true changes in CTC counts. After sample processing and imaging, CTCs are identified in 140 four-channel immunofluorescence images. This process starts with the identification of events of interest in the images through segmentation [7]. Based on this segmentation, CTC candidates are selected and the corresponding thumbnail images are presented to the user. Ideally, the task of segmentation is to identify every single event of interest inside a sample. Segmentation should not join multiple events together, nor split single events into multiple ones. As cell density increases, the segmentation task becomes more challenging. In samples derived from diagnostic leukapheresis (DLA) and run on the CellSearch system [8,9] we noticed that in these cell dense samples the segmentation used by CellSearch failed to identify single objects, leading to artificially low CTC counts. A better segmentation was achieved by the open-source image analysis program ACCEPT [10]. However, ACCEPT employs an active contour method for segmentation and joins objects that are in contact with each other into a single event, thus failing to identify single objects in denser samples. StarDist is a Deep Learning based method that describes all events with a star-convex shape and which is quite effective in segmenting single cells in tissue sections [11]. Here, we evaluate StarDist for the segmentation of all objects as single events in CellSearch images corresponding to peripheral blood and DLA samples. Using StarDist, we demonstrate a clear improvement in the segmentation of CellSearch images leading to a more accurate CTC count.

## 2. Materials and Methods

### 2.1. Sample Archives to Evaluate

For the comparison of CellSearch and StarDist segmentations, 1163 CellSearch archives of previously scored CellSearch images were used. This set included peripheral blood samples of 90 healthy donors (NCT00133913 [12]), 442 castration-resistant prostate cancer patients (NCT00133900 [13]), and 601 DLA samples of prostate (*n* = 24), breast (*n* = 49), and non-small cell lung cancer patients (*n* = 528) [8,14,15]. The time between sample collection and sample preparation was known for 436 whole blood prostate cancer samples and for 322 of the DLA non-small cell lung samples. All study participants had signed informed consent forms per the Helsinki declaration and all protocols were approved by the Ethics Committees of the respective studies.

### 2.2. Stardist Segmentation

The StarDist method from github (https://github.com/stardist/stardist, accessed on 1 June 2021) was applied with a few non-default parameters. Specifically, for network “n_channel_in” was set to 3 and “u_net_n_depth” was set to 4, for optimization “train_background_reg” was set to 0.0004, “train_learning_rate” was set to 0.001 and train_batch_size” was set to 8, and for post-processing “nms_thresh” was set to 0.3 and “prob_thresh” was set to 0.3. The input images were locally contrast-enhanced. Training data were generated in QuPath 0.2.3 [16] using the procedure described in the StarDist documentation.

### 2.3. Performance in High Density Samples

The cell density of CellSearch samples varies greatly from sample to sample, with occasional high densities in blood samples and more often in DLA. To assess the performance of StarDist in higher density samples, we created samples with increasing amounts of magnetically labeled white blood cells and a fixed amount of tumor cells. For this, we magnetically labeled, separated, and fluorescently stained white blood cells from blood using the CellSearch system with the standard reagents, except for the EpCAM-ferrofluid, which was replaced by streptavidin ferrofluid (Biomagnetic solutions, State College, PA, USA) coupled to CD45-biotin. Cells of the prostate cancer cell line LNCaP were processed using the regular CellSearch reagents. The cell concentrations of the resulting samples were determined and an increasing number of white blood cells together with ~6000 cells from the prostate cancer cell line LNCaP were spiked into a mixture of PBS and CS-fixative to reach a final volume of 325 µL. The spiked samples were manually placed in CellSearch cartridges and scanned using the CellTracks Analyzer II system.

In our experience, an increase in the time between blood draw and sample prep causes a higher white blood cell carry-over during enrichment and therefore a higher cell density in the resulting samples, while as mentioned DLA samples show an even higher cell density. To test this assumption, we evaluated the number of segmented events by StarDist using 436 whole blood and 322 DLA samples for which the time between sample collection and sample prep was known.

### 2.4. Recovery of CellSearch CTC by StarDist and Potential Gain in CTC

As CellSearch is the gold standard in the enumeration of CTC, we compared the StarDist segmentations to those from CellSearch. The CellSearch segmentation draws rectangular segmentations with a 10-pixel margin around the event, whereas StarDist draws a star-convex segmentation without any margin. The vast majority of StarDist segmented events are cytokeratin-PE negative white blood cells, whereas CellSearch only segments events that express some cytokeratin-PE and some DAPI. A direct comparison of segmentations is therefore not meaningful. To ensure that the StarDist algorithm does not cause any losses of events that were scored by a reviewer as CTC in CellSearch, we selected CellSearch segmentations in which a CTC was detected (with a maximum of 20 from a single sample). Together with the corresponding StarDist segmentation, these were shown to a reviewer, who evaluated whether (1) the CTC found by CellSearch were segmented by StarDist, and (2) whether the segmentations were correct. A correct segmentation outlines the whole event and only the event, so it does not split a single event into two segmentations and it does not join multiple events together. For CTC enumeration by a human reviewer, a CTC just needs to be included in the segmentation; however, for automated enumerations, a correct outline is also important.

To assess whether StarDist segments any CTC that were not detected by CellSearch, we presented to four reviewers all events detected in 442 blood samples as well as seven DLA samples that were CTC candidates based on their staining properties, but which were not selected by the CellSearch segmentation and therefore had not been evaluated by the original reviewer. For this purpose, a possible CTC was defined as an event meeting all of the following requirements: cytokeratin-PE intensity maximum >75 and mean >45, DAPI intensity maximum >60 and mean >50, CD45-APC mean intensity <50 or mean PE intensity >1000 (because some crosstalk exists from PE to the APC channel), and a total event area between 36 and 1000 pixels, with a stained area in PE of at least 30 pixels. These boundaries were loosely based on an existing ACCEPT CTC definition [17]. We do not expect that this definition will include all CTC, and we know that the majority of events included by this definition are not CTC. To assess if the more sensitive StarDist segmentation would lead to an increase in false positives, we also performed this selection and evaluation for 93 healthy donor samples for which the CellSearch results were reported previously [9]. 

### 2.5. Code Environment

All trainings and evaluations were performed in Python 3.7, utilizing StarDist 0.6.2 [11], Pandas 1.1.5 [18], Numpy 1.18.5 [19], OpenCV 4.0.1 [20], Matplotlib 3.3.4 [21], scipy 1.4.1 [22], tifffile 2021.3.31 and skimage 0.18.1 [23] packages and associated dependencies.

## 3. Results

### 3.1. Performance of Segmentation Algorithms on Dense CellSearch Images

Segmentation algorithms developed for CellSearch images include the built-in algorithm and the active contour method employed by ACCEPT. However, both the ACCEPT as well as the CellSearch segmentation methods were developed for CTC samples in which the cell density is such that adjacent cells do not touch each other. In high cell density samples, including most DLA samples and some whole blood samples, the segmentation algorithms join all cells that are in contact with each other into a single event, as shown in Figure 1. In addition, CellSearch seems to miss cells that should have been presented to a reviewer. In contrast, StarDist achieved segmentation of almost all single events, as illustrated in Figure 1. An overview of the capabilities of the different methods is given in Table 1. 

### 3.2. Extent of the Problem in CellSearch Segmentation

In the CellSearch segmentation algorithm, a single cell of 20 µm diameter will result in a segmentation of 2500 pixels. Therefore, we reasoned that those segmentations larger than 2500 pixels, the equivalent of 32 × 32 µm^2^, are likely to contain more than one cell. For these sizes, it becomes probable that multiple CTCs are contained within a single segmentation, leading to an undercount of CTCs. Furthermore, when the segmentations get even larger, the review becomes more difficult due to presentation issues, likely leading to misclassifications. Presentation issues become noticeable when segmentations are around 70 × 70 µm^2^ (equivalent to 11,881 pixels) and become much worse for larger segmentations. The occurrence of ‘regular’ (≤2500 pixels), larger than a cell (2500 to 25,000 pixels), too large to review (25,000–100,000), and very large segmentations (>100,000 pixels), as well as the area of the cartridge and the segmented area consisting of these segmentation sizes, are listed in Table 2.

Overall, in whole blood of cancer patients, 40% of segmentations are larger than a large single cell, and 0.2% of segmentations are so large that review is possibly affected. In whole blood of healthy donors, 35% of segmentations are larger than a large single cell, and 0.07% of segmentations are so large that review is possibly affected. Very large segmentations of more than 100,000 pixels were observed in 9% (55 of 601) of DLA samples, comprising up to 70% of the segmented area. One out of 442 whole blood samples from cancer patients also had very large segmentations, which comprised 0.7% of the segmented area. For these cartridges, the accuracy of the CTC count is compromised. To illustrate these sizes, Figure 2 shows an example for a segmentation of approximately 2500, 25,000, and 160,000 pixels.

### 3.3. Cell Density 

To assess the performance of StarDist in cartridges with various cell densities, spike-in experiments were performed. The number of cells spiked was approximately 6000 LNCaP cancer cells together with zero, 50-, 100-, 150-, 200-, 300- and 400-thousand white blood cells. In a CellSearch cartridge and scan, 400,000 cells correspond to ~3000 cells per image and about 50% of the surface area covered by cells. Larger cell concentrations are not evaluable as the CellTracks Analyzer II fails to perform autofocus in denser samples. Figure 3 shows the result of these spike-in experiments. It can be seen that at very low concentrations, StarDist detects an excess of approximately 30.000, mostly small, events, while a good correlation is seen up to the maximal density. Although useful to evaluate the ability of StarDist to segment high-density samples, these titrations do not contain the clumped and broken cells often seen in high-density DLA samples.

To assess the dependence between white blood cell carry over, causing high cartridge densities, and the time between sample collection and sample preparation, we calculated the number of events in 436 blood as well as 322 DLA samples. In Figure 4 the empirical cumulative distribution functions (CDF) for the number of StarDist events per sample is shown for blood as well as DLA samples with one, two, three, and four days between sample collection and sample prep. There is a large variation in the number of events per sample and it can be seen that for blood the total number of events increases when there is more than one day between the sample collection and sample prep. In fact, for whole blood, there is first-order stochastic dominance of the empirical CDF for one day between collection and prep and the empirical CDFs for two, three, and four or more days. The median number of events for whole blood samples with one day between collection and prep was 13-thousand, in contrast to 28-, 37- and 54-thousand events for two, three, and four or more days respectively. DLA samples have a much less pronounced dependence on the time between sample collection and sample prep. For DLA samples, the median number of events was 108-, 136-, 138- and 156-thousand for one, two, three, and four or more days respectively.

### 3.4. Recovery of CellSearch CTC by StarDist and Potential Gain in CTC

We evaluated 1948 CTC identified by reviewers after CellSearch segmentation to verify that these CTC were also segmented by StarDist and assessed whether the segmentation was within a few pixels of the true outline. We found that one CTC was not segmented by StarDist, resulting in a StarDist segmentation of 99.95% of CTC segmented by CellSearch. Of the segmentations, 14 (0.7%) were split into multiple events, while 8 (0.4%) segmentations included only part of the event. In 11 (0.6%) segmentations, the CTC was merged with (part of) an adjacent object into a single event. Such splitting and merging errors are not problematic for a human reviewer, but may cause automated enumeration methods to exclude such events, leading to a lower recovery of CTC. 

To determine the extent to which CTCs were missed by the CellSearch segmentation, we reviewed a total of 36,996 events from 442 whole blood image archives that were CTC candidates based on their staining properties, but had not been reviewed in the original review because they were not segmented by CellSearch. Of these events, 762 were CTC by consensus of a panel of four reviewers. Ordinary least-squares linear regression showed an average increase of 8.7% of the number of CTC found in the original review plus 0.214 CTC (CTC_StarDist_ = 1.087 × CTC_CellSearch_ + 0.214, R^2^ = 0.99). Of the 218 samples that originally had zero CTC, 20 samples (9.2%) gained at least one CTC after implementing StarDist segmentation and CTC review. In two of the 442 samples, the increase in CTC led to a conversion from a ‘lower risk’ (<5 CTC) to ‘a higher risk’ (≥5 CTC) group, see Figure 5. Additionally, in seven image archives of prostate cancer DLA samples we reviewed a total of 4666 events that were CTC candidates based on their staining properties, but were not reviewed because they were not segmented by CellSearch. Of these events, 219 were CTC by consensus of the reviewers. Ordinary least squares linear regression showed an average increase of 19.8% of the number of CTC found in the original review plus 1.136 CTC (CTC_StarDist_ = 1.198 × CTC_CellSearch_ + 1.136, R^2^ = 0.99).

In image archives from 90 healthy donor samples, two events segmented by StarDist but not by CellSearch were selected by the reviewers as CTC. Human reviewers had previously identified three CTC in these samples. Taken together, 5.4% of the healthy donors had one (false positive) CTC, while none had two or more. This percentage is in line with the 5.5% of healthy donor samples containing one CTC-like event as found in the original CellSearch study [24] and does not indicate a large number of false positives as a result of the segmentation.

As a sample archive consists of a number of adjacent images, some events will be present on the image edges. The segmentation of events on an edge is more challenging for segmentation algorithms that employ pixels surrounding the event, for example in the local contrast enhancement employed in CellSearch. Additionally, the non-uniformity of the sample illumination means events near the edge have lower signal to noise, which could reduce the likelihood of being segmented. To assess if the events missed by CellSearch are predominantly present on the edges of the image, we created a heat map consisting of 25 × 31 bins (~25 × 25 µm per bin) displaying the locations of the CTC segmented only by StarDist. In the heatmap, as shown in Figure 6, it can be seen that although additional events are found throughout the image, there is a higher incidence at the edges, with 36.6% of events being present in an edge bin, compared to 14.5% expected for a uniform distribution. The highest concentration of missed events is on the left and top edges, which could be attributed to the implementation details of the CellSearch algorithm.

## 4. Discussion

For the classification of objects in fluorescence images, single objects need to be identified and presented to a reviewer, an algorithm, or a combination of both. During the development of the CellSearch system, an algorithm segmenting image sections containing both DAPI and CK-PE staining was chosen to facilitate CTC enumeration. These sections are presented to a reviewer who decides which ones contain a CTC. Failure to segment a CTC means this CTC will not be presented to the reviewer, and thus, not accounted for in the final CTC count. Furthermore, incorrect segmentations can contain two or even more clearly separated CTC. Another possibility is that the segmentations are too large to be reviewed effectively, as they can encompass a whole image or more. Such failures can also lead to underestimation of the real CTC load, albeit less dramatically than when the CTC is not presented at all. 

Here, we evaluated StarDist as an alternative to the CellSearch segmentation and demonstrated that it would lead to the segmentation of on average 8.7% + 0.21 more CTC in whole blood, while barely missing any (0.05%) of the CTC detected using the CellSearch segmentation. In DLA samples, the advantage of StarDist is even more profound, since we detected on average 19.8% + 1.14 more CTC, albeit in a small sample size of only seven DLA samples. For the current evaluation, a pre-selection of CTC candidates was made that likely missed some CTC. The average 10.5% CTC gain found with this selection thus represents a lower bound for the number of CTC not segmented by the original CellSearch segmentation. Using a deep learning classification algorithm currently in development [25], we estimate the upper bound of the average CTC gain due to improved segmentation to be 38%. Considering the large number of events segmented per sample (mean 54 × 10^3^) using the StarDist method, a review of all segmented events is neither feasible nor meaningful. If StarDist was applied in the current CellSearch workflow, a pre-selection of events to be presented to the reviewer would be needed to reduce the number of candidate CTC. Fully automated classification could be performed using a fully designed gating strategy as applied in ACCEPT [17], or through a deep learning approach as presented previously [25]. The main advantage of the ACCEPT approach is that it is relatively simple and thus interpretable. The main downside is that it has a limited ability to encode more complex classification rules. A deep learning approach would be able to encode very complex classification rules, and the major downside is that these classification rules are typically not interpretable. While deep learning classifiers have been shown to outperform fully designed classifiers in many classification tasks, deep learning classifiers require a large number of annotated segmented image sections before they are reliable and can be quirky when trained on insufficient data. For the identification of clusters, it is expected that regardless of the used method also the properties of the surrounding events will need to be taken into account during classification, as StarDist segments adjacent cells into separate segmentations. 

Recent data revealed that the enumeration of tdEV in the original CellSearch images further improves the prognostic stratification of CTC [26]. The CellSearch segmentation algorithm however does not present the tdEV for manual review. This shortcoming was overcome by the introduction of ACCEPT which allows gating and enumeration of tdEVs [26]. ACCEPT employs a Bregman active contour method that finds the outline of each event. The event outline permits quantitative characterization of the event in terms of signal intensity (e.g., mean intensity and max intensity) and morphology (e.g., area and eccentricity). These extracted values are useful for the identification of CTC [17] as well as for the quantification of marker expression on the CTC [27] which can for instance be used to identify epithelial to mesenchymal transition (EMT). We expect that this identification and characterization will also be possible with StarDist segmentations as they are sufficiently accurate event outlines, and the segmented events include tdEV, bare nuclei, and white blood cells. Furthermore, in high cell densities, as found in some whole blood samples and most DLA samples, StarDist continues to perform well while ACCEPT fails to find the outline of single events. 

The StarDist network was optimized to detect all cell types as well as tdEVs in the CellSearch archives, because we wish to investigate the prognostic potential of all sample constituents. Another possible approach for CellSearch samples would be to train two networks separately for the identification of CTC and tdEV. This may be a more performant approach if the aim is only to enumerate CTC or tdEV. However, the current training allowed us to look at the impact of sample age on the total number of cells in a cartridge. Here we found that for whole blood the number of events in a cartridge is on average more than two-fold higher when more than one day has passed between sample collection and preparation. No relationship was observed between sample age and the total number of CTC nor on the number of CTC segmented only by StarDist. This suggests non-specific binding for white blood cells is increased in older samples. For DLA samples, a similar, albeit relatively smaller, effect could be observed.

The star-convex model that is applied by StarDist also has its limitations, some examples of which are shown in Figure 7A–C. Panel A shows two cells that are difficult to describe with a star-convex model, and StarDist splits these cells into multiple segmentations. Such cells are relatively rare in CTC samples, but do mean that StarDist could not be applied for the segmentation of circulating endothelial samples. Furthermore, Panel B shows very faint events that are close to bright events. These are false negative events in both StarDist and CellSearch. This behavior could be improved for StarDist by reducing the local window size used in pre-processing, but this would result in false-negatives for areas with densely packed cells such as CTC clusters. Panel C shows the plastic edge of a CellSearch cartridge, where StarDist segments small variations in autofluorescence. 

In the aforementioned cases, the true segmentation can be easily identified by the human reviewer, but there are also instances in which the true segmentation is difficult to ascertain. Panel D in Figure 7 shows some examples of events where StarDist does perform a segmentation, but the human reviewer cannot determine whether this segmentation is correct or not.

## 5. Conclusions

Here, we demonstrated the occurrence of critical failures in samples segmented by the built-in CellSearch segmentation algorithm. The ACCEPT algorithm developed for CellSearch samples did not perform well in dense (DLA) samples, because it joined nearby objects together into a single event. To overcome this we evaluated the StarDist deep-learning-based segmentation method and found it outperforms both ACCEPT and the current CellSearch segmentation in both whole blood as well as DLA samples. The StarDist method segments individual outlines up to the maximal cell density that can be scanned using the CellTracks system while also segmenting tdEV. The StarDist segmentations closely follow the cell outline in most cases, enabling precise quantification of signal intensities. These intensities subsequently enable quantitative phenotypic characterization of the segmented events. Furthermore, we also found that StarDist segmented at least an additional 10% of CTC in CellSearch whole blood samples, and an additional 20% of CTC in CellSearch DLA samples, while recovering 99.95% of all CellSearch selected CTC.

## Figures and Tables

**Figure 1 cancers-14-02916-f001:**
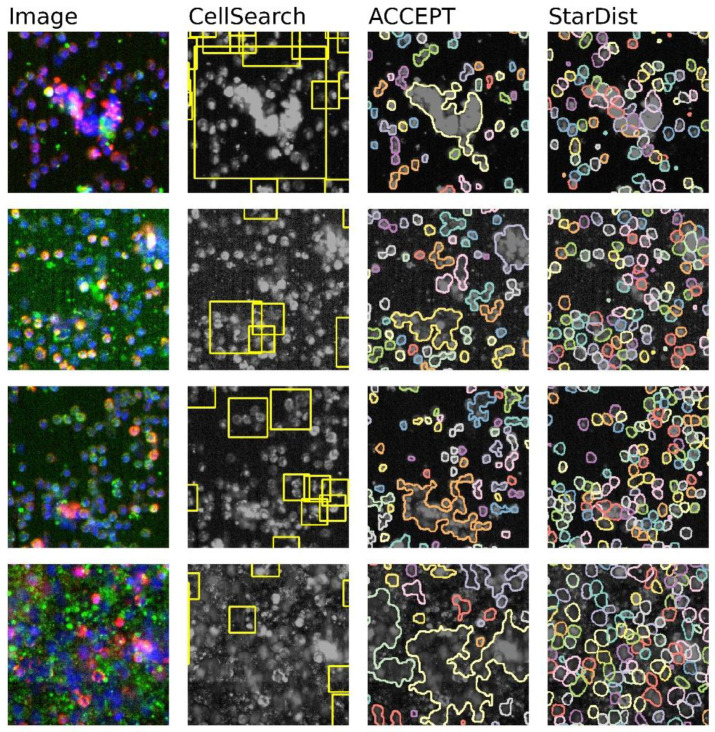
Four example images with the accompanying segmentations as performed by CellSearch, ACCEPT, and StarDist. The first column shows a false-color image in which the nuclear staining is colored blue (DAPI), the cytokeratin staining green (CK-PE), and CD45 staining red (CD45-APC). The second column shows the same image in black and white with the yellow rectangular CellSearch segmentation areas. In the third and fourth columns, the ACCEPT and StarDist segmentations are depicted as a separate color for each segmented event. StarDist clearly performs best in segmenting all single events separately, especially in areas containing a high cell density.

**Figure 2 cancers-14-02916-f002:**
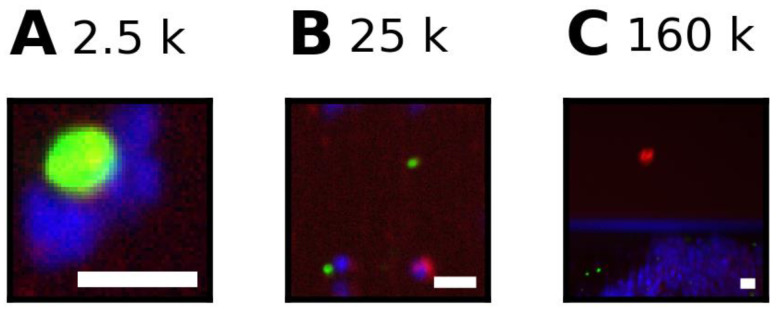
Examples of different segmentations sizes (**A**) Segmentation of 2500 pixels, (**B**) segmentation of 25,000 pixels, and (**C**) segmentation of 160,000 pixels.

**Figure 3 cancers-14-02916-f003:**
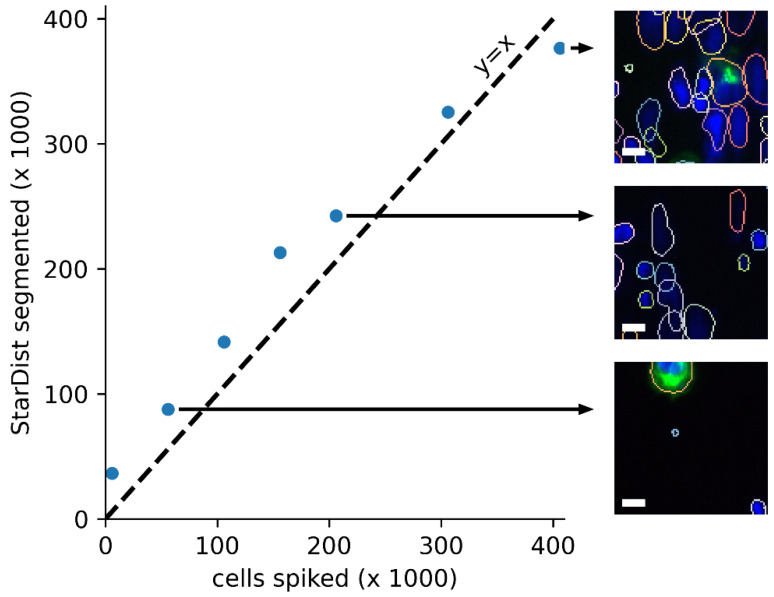
Performance of StarDist segmentation on increasing numbers of spiked cells. Comparison of number of spiked cells in the cartridge versus the number of events detected by Stardist showing a good correlation up to the maximal density that can be imaged using the CellTracks Analyzer II (scale bar is 10 µm).

**Figure 4 cancers-14-02916-f004:**
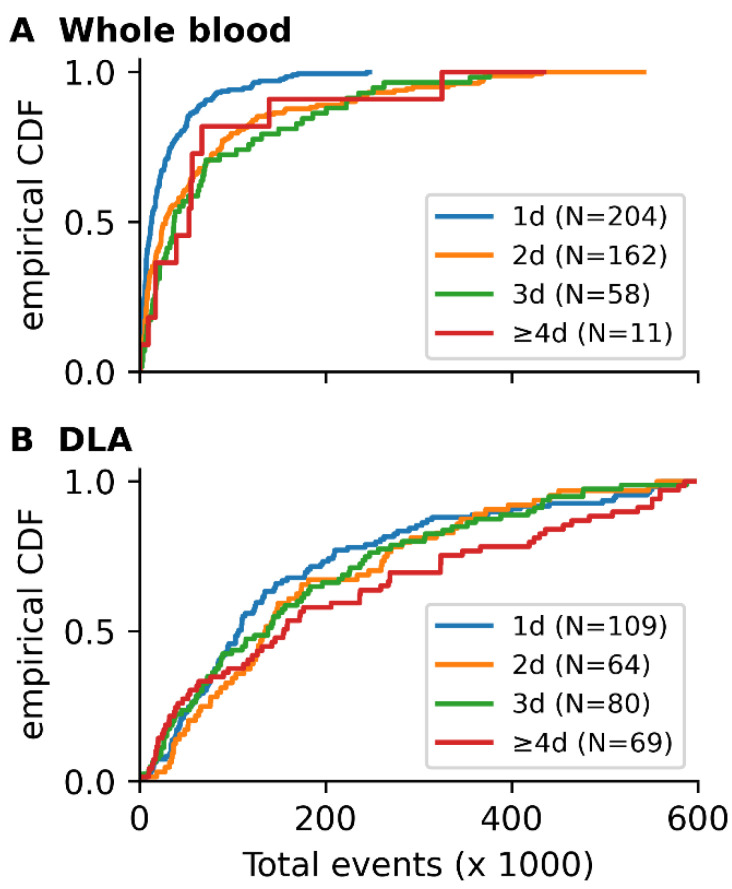
Cumulative distribution functions (CDF) of (A) blood and (B) DLA samples with different collection to prep intervals. (**A**) The cumulative distribution functions of samples with one, two, three, and four+ days (d) between blood draw and sample preparation show a clear increase in cell density for time intervals of more than one day in (**A**) whole blood and (**B**) DLA samples. While a dependence of the number of events on the time from sample collection to sample preparation may also be observed in DLA samples, the difference is less pronounced than in blood.

**Figure 5 cancers-14-02916-f005:**
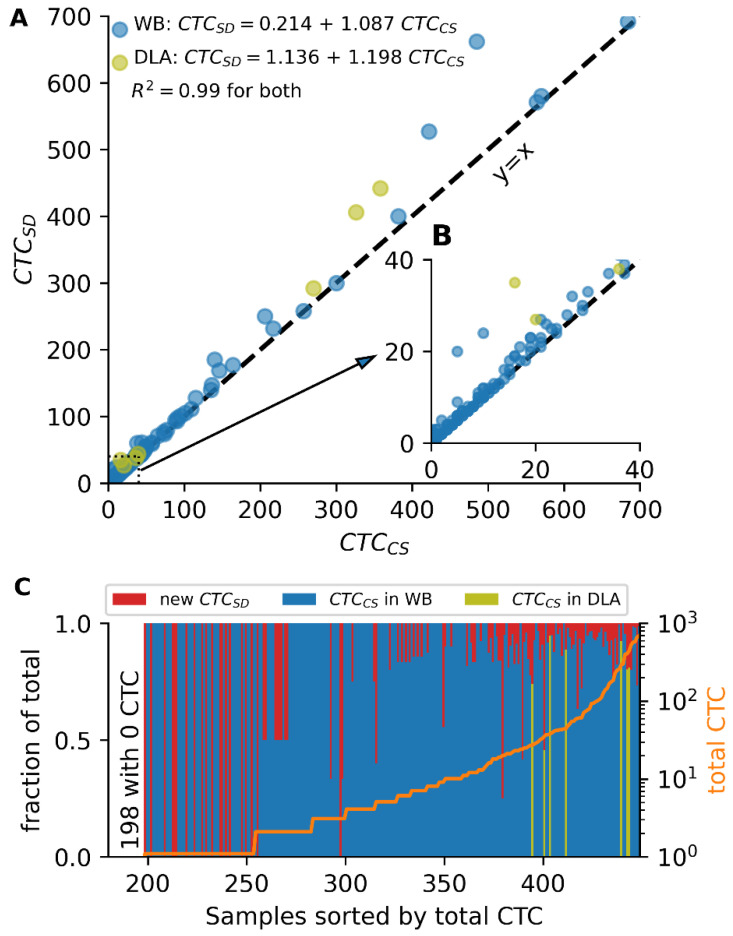
CTC found by CellSearch and StarDist segmentation in archive images of whole blood and DLA. (**A**) Comparison of the original number of CTC found by CellSearch and the combined number of CTC found after evaluation of additional StarDist segmentation for all samples. (**B**) Zoom in of A showing the samples with <40 CTC only. (**C**) Fraction of total CTC found by original review and StarDist review for each sample. Samples are sorted on the total number of CTC as depicted by the orange line. The first 198 samples have no detected CTC and therefore have no fractions.

**Figure 6 cancers-14-02916-f006:**
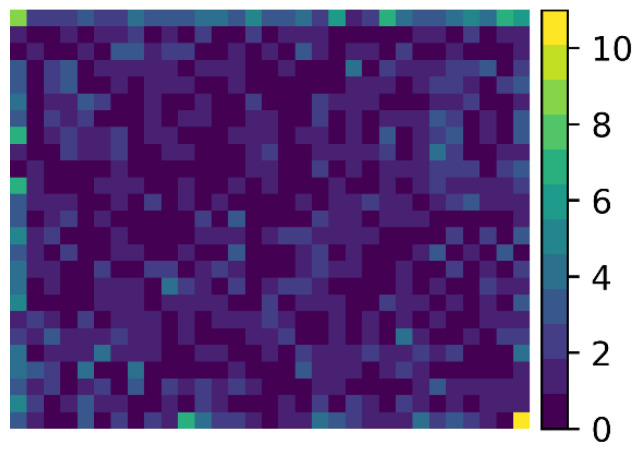
Heatmap of the image location for the additional CTC segmented by StarDist. Heat map of 25 × 31 bins displaying the number of additional CTC segmented by StarDist for each bin. Heat map shows an increased concentration of events on the left and top edges of the image.

**Figure 7 cancers-14-02916-f007:**
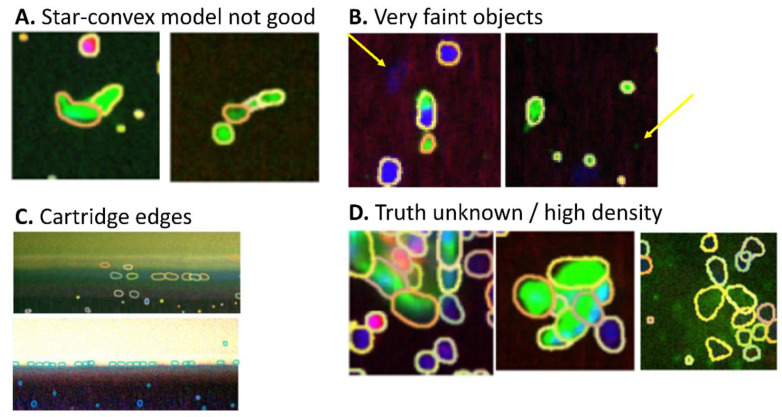
Limitations of StarDist segmentations and unknown truth. (**A**) Non-star-convex shapes cannot be segmented properly by StarDist. (**B**) Some very faint objects are not segmented. (**C**) Cartridge edges can lead to segmentation artifacts in StarDist. (**D**) Examples of images where the ‘true’ segmentation is unknown.

**Table 1 cancers-14-02916-t001:** Suitability of CellSearch, ACCEPT, and StarDist for segmentation tasks. Overview of the suitability of the CellSearch, ACCEPT and StarDist for the pre-selection of likely CTC, identification of cell outlines, segmentation of cells in high-density samples, and ability to resolve cells that are part of a cluster.

Segmentation Method	CTC Pre-Selection	Cell Outlines	High Cell Density	Segments Cells inside Clusters
CellSearch	YES	NO	NO	NO
ACCEPT	YES	YES	NO	NO
StarDist	YES	YES	YES	YES

**Table 2 cancers-14-02916-t002:** CellSearch segmentation sizes in blood and DLA samples. The percentage of samples with different segmentation sizes present in whole blood and DLA samples, the mean percentage of the total cartridge area covered, and the mean percentage of the total segmented area by the different CellSearch segmentation sizes. The ranges are shown in parentheses.

	Diagnostic LeukApheresis		
Segmentation Size (Pixels)	% of Samples	% of Cartridge Area	% of Area Segmented
<2500	100	3.43 (0.00–38.32)	54.77 (7.79–100)
2500–25,000	99.83	3.10 (0.00–35.83)	43.11 (0.00–84.15)
25,000–100,000	38.94	0.31 (0.00–11.28)	0.95 (0.00–14.19)
>100,000	9.15	0.83 (0.00–51.78)	1.17 (0.00–69.81)

## Data Availability

The data presented in this study are available on request from the corresponding author.

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
