# Peer review of "StarDist Image Segmentation Improves Circulating Tumor Cell Detection"

_cancers, 2022, doi:10.3390/cancers14122916_

Round 1

Reviewer 1 Report

The manuscript was prepared very well. The introduction section justifies the purpose of the study. I congratulate the authors for the preparation of the manuscript

However, I have the following comments:

Introduction

·       “Accurate enumeration of CTC is important for a correct patient risk assessment and to evaluate treatment response or disease progression during the course of the disease by true changes in CTC counts” Include appropriate reference 10.3390/diagnostics10040215.

·       Include background on the use of the CellSearch in different types of cancer and its clinical potential. 10.3390/diagnostics10070443.

·       Indicate why you need to use your new diagnostic platform

Materials and Methods

·       The methodology is perfectly described

Results

·       The tables/figures and the text describing them do not require any input, it is the strongest part of this study.

Discussion

·       What is new in this manuscript for diagnosing cancer patients?

·       Include a section on limitations and strengths.

·       What does this article contribute to, the authors should make their own assessment and include their own discussion of the results shown in the manuscript?

Reviewer 2 Report

Recommendation: Publish after major revisions noted.  

Comments:  

This manuscript describes StarDist image segmentation to circulating tumor cell detection. The authors need to address the following comments and revise the manuscript accordingly.   

  1. This manuscript is in need of substantial editing, English language and style improvement. 
  2. Introduction: Please consider to add a paragraph describing circulating biomarkers, and the following references. a) Roy, D.; Pascher, A.; Juratli, M.A.; Sporn, J.C. The Potential of Aptamer-Mediated Liquid Biopsy for Early Detection of Cancer. Int. J. Mol. Sci. 2021, 22, 5601. https://doi.org/10.3390/ijms22115601 b) Toss, A., Mu, Z., Fernandez, S., & Cristofanilli, M. (2014). CTC enumeration and characterization: moving toward personalized medicine. Annals of translational medicine, 2(11), 108. https://doi.org/10.3978/j.issn.2305-5839.2014.09.06 c) Roy D, Lucci A, Ignatiadis M, Jeffrey SS. Cell-free circulating tumor DNA profiling in cancer management. Trends Mol Med. 2021 Jul 23:S1471-4914(21)00182-9. doi: 10.1016/j.molmed.2021.07.001. d) Vogl TJ, Riegelbauer LJ, Oppermann E, Kostantin M, Ackermann H, Trzmiel A, et al. (2021) Early dynamic changes in circulating tumor cells and prognostic relevance following interventional radiological treatments in patients with hepatocellular carcinoma. PLoS ONE 16(2): e0246527. https://doi.org/10.1371/journal.pone.0246527.
  3. Page 4, line 140: Provide a table for CellSearch, ACCEPT and StarDist and show their advantages and limitations.
  4. Page 6, line 195: Cumulative distribution functions (CDF)
  5. Page 9, Fig 6: Indicate the color scale with unit.
  6. Please consider adding a paragraph on the model fitting for the clusters. Provide a side-by-side comparison with examples.
  7. Discuss the potential of the model for epithelial to mesenchymal transition (EMT) markers.
  8. Address model overfitting and cell delineation. 
  9. Consider to highlight contextual features and inspection of neighboring structures.
  10. Consider providing a table detailing the influence of characteristics on classifier accuracy.

Round 2

Reviewer 2 Report

Publish it.